# Fatty Acid Composition of Cosmetic Argan Oil: Provenience and Authenticity Criteria

**DOI:** 10.3390/molecules25184080

**Published:** 2020-09-07

**Authors:** Milena Bučar Miklavčič, Fouad Taous, Vasilij Valenčič, Tibari Elghali, Maja Podgornik, Lidija Strojnik, Nives Ogrinc

**Affiliations:** 1Science and Research Centre Koper, Institute for Olive Culture, 6000 Koper, Slovenia; milena.bucarmiklavcic@zrs-kp.si (M.B.M.); vasilij.valencic@zrs-kp.si (V.V.); maja.podgornik@zrs-kp.si (M.P.); 2Centre National De L’énergie, Des Sciences Et Techniques Nucleaires, Rabat 10001, Morocco; taous@cnesten.org.ma (F.T.); elghali@cnesten.org.ma (T.E.); 3Department of Environmental Sciences, Jožef Stefan Institute, Jamova cesta 39, 1000 Ljubljana, Slovenia; lidija.strojnik@ijs.si

**Keywords:** Argan oil, fatty acids, geographical origin, Morocco

## Abstract

In this work, fatty-acid profiles, including trans fatty acids, in combination with chemometric tools, were applied as a determinant of purity (i.e., adulteration) and provenance (i.e., geographical origin) of cosmetic grade argan oil collected from different regions of Morocco in 2017. The fatty acid profiles obtained by gas chromatography (GC) showed that oleic acid (C18:1) is the most abundant fatty acid, followed by linoleic acid (C18:2) and palmitic acid (C16:0). The content of trans-oleic and trans-linoleic isomers was between 0.02% and 0.03%, while trans-linolenic isomers were between 0.06% and 0.09%. Discriminant analysis (DA) and orthogonal projection to latent structure—discriminant analysis (OPLS-DA) were performed to discriminate between argan oils from Essaouira, Taroudant, Tiznit, Chtouka-Aït Baha and Sidi Ifni. The correct classification rate was highest for argan oil from the Chtouka-Aït Baha province (90.0%) and the lowest for oils from the Sidi Ifni province (14.3%), with an overall correct classification rate of 51.6%. Pairwise comparison using OPLS-DA could predictably differentiate (≥0.92) between the geographical regions with the levels of stearic (C18:0) and arachidic (C20:0) fatty acids accounting for most of the variance. This study shows the feasibility of implementing authenticity criteria for argan oils by including limit values for trans-fatty acids and the ability to discern provenance using fatty acid profiling.

## 1. Introduction

Argan oil is produced from the kernels of the argan tree (*Argania spinosa* (L.) Skeels) that is common to Morocco and parts of Algeria. As a multi-purpose tree, it plays a vital socioeconomic role, and its sustainable exploitation is essential in maintaining an ecological balance and preserving biodiversity [1]. Although the argan tree is the only representative species of the tropical family *Sapotaceae* in Morocco, it is the second-largest forest species after oak and before cedar and has a life span of 200 years or more. It is grown mainly in the arid and semi-arid regions of southwestern Morocco covering about 870,000 ha. The estimated twenty million trees are spread over the provinces of Essaouira (130,000 ha), Agadir-Ida ou Tanane (37,000 ha), Chtouka-Aït Baha (90,000 ha), Tiznit (140,000 ha), Taroudant (360,000 ha) and Inzeguane-Aït Melloul (13,000 ha), representing just about 17% of the original Moroccan forest area. The importance of the argan tree was recognised by UNESCO in 1998, who declared it a “protected species” and the extensive intra-montane plain, which is home to the Argan tree, a biosphere reserve [2]. The argan tree has been introduced as a cultivated species in the deserts of Tunisia, Israel and South Africa, among other parts of the world [3]. Argan oil’s significance in Moroccan culture derives from its traditional use in local cuisine, medicine and cosmetics. The global argan oil market is expected to grow at a compound annual growth rate of 10.8% from 2019 to 2027 to reach USD 507.2 million by 2027 [4]. The income generated creates a livelihood that sustains and supports many households and rural communities that are dependent upon the Arganeraie Biosphere Reserve [2,3]. 

Khallouki et al. [1] best described the appearance of the argan fruit as “a false drupe, oval sessile bay, fusiform, rounded or in a spindle and is about 4 cm long with a very hard nut containing two or three kernels representing about a quarter of the weight of the fresh fruit”. Once ripe, the nuts are collected by rural women and, to a lesser extent, men living within the reserve. The women are also the ones who practice the traditional manual methods of preparing the nuts to extract the oil. The nuts are first broken with rocks to remove the kernels, which are then air-dried in clay containers and roasted in the case of cooking oil production but left unroasted for producing cosmetic grade oil. In the final stage, the women grind and press (cold pressing) the kernels by hand to release the oil. Women’s co-operatives now do much of the work, and the kernels are processed mechanically using a scraper-pulping machine and the oil extracted by mechanical cold pressing [3]. Charrouf and Guillaume [3] developed the concept of these women’s co-operatives and remains the leading figure in the valorisation of argan oil in Morocco. Mechanical processing considerably reduces production time, increases yield (>45% *v/w*) and guarantees consistent extraction of high-quality oil [1]. 

In 2010, argan oil received the Protected Geographical Indication (PGI) recognition by the Moroccan Government, which means the term “*argan oil*” can only be used to describe oil whose production is closely linked to a specific geographical area with at least one of the stages of production, processing or preparation taking place in that area. In addition, because Morocco is the only country that produces argan oil, it has become vital to establish the methodologies that can verify the oil’s authenticity and provenance. To date, several techniques have been applied to check the quality and possible adulteration of argan oil. For example, Hilali et al. [5] proposed campesterol as a suitable marker of oil purity, since it is present in much lower concentrations in argan oil than in most vegetable oils. In this way, the authors assessed argan oil purity “*unambiguously up to 98%*” [5]. Salghi et al. [6] developed a method based on triacylglycerol profiles that were determined using high-performance liquid chromatography with evaporative light scattering detection (HPLC-ELSD) for assessing the adulteration of argan oil with sunflower, soybean and olive oil up to the level of 5%. Alternatively, Mohammed et al. [7] proposed using dietary elements to detect adulteration with some success. They used inductively coupled plasma atomic emission spectroscopy (ICP-AES) to determine the levels of eight elements (Cd, Cr, Cu, Zn, Fe, K, Mg and Ca). Their main conclusion was that processing of the argan kernels did not significantly modify the elemental composition, which could be used to detect adulteration, in addition to measuring the campesterol content. 

In contrast, few reported studies attempt to verify the provenance of argan oil. Recently, the group led by Fouad investigated the use of stable isotope ratios of carbon (^13^C/^12^C) and oxygen (^18^O/^16^O) to authenticate argan oil from different Moroccan co-operatives (PGI certified) from five provenances in the southwest part of Morocco [8]. Although their study shows that the use of stable isotopes is an efficient tool for the verification of provenance, other analytical variables such as multi-element profiles, spectroscopic fingerprinting and chemical profiling are needed to improve classification accuracy [9]. Fatty acid profiling, when combined with various chemometric tools, has been successfully applied to discriminate the geographical origin of vegetable oils including olive oil [10,11], pumpkin oil [10,12] and *Sativa* oils [13]. 

In the case of argan oil, Belcadi-Haloui et al. [14] compared the fatty acid profile of sample oils from three regions in the southwest of Morocco: Aït Melloul, Argana and Tamanar. The authors found that the concentration of saturated fatty acids (palmitic and stearic acid) was similar in oils from all three regions, but they did note that there were significant differences in the concentrations of unsaturated oleic and linoleic acids. The study, however, only noted differences and did not focus on determining geographical origin. Kharbach et al. [15], who analysed 150 samples of extra virgin argan oil (EVAO) with PGI from five regions of the Moroccan argan forest (Aït-Baha, Agadir, Essaouira, Tiznit and Taroudant), reported similar results [16]. They found that the fatty acid composition was dependent on the oil’s geographic origin; however, PCA showed significant overlapping of Agadir, Essaouira, Ait-Baha and Tiznit oils. With further data processing, the authors were able to create two distinct groups, which represent overlays between oils from Agadir and Essaouira and oils from Ait-Baha, Taroudant and Tiznit. However, the authors were able to classify EVAO according to its geographical origin and, in some cases preparation method, using a combination of UV-Visible fingerprinting and chemometric tools. Aithammou et al. [17] studied the effect of clones, year of harvest and geographical origin of fruits on the quality and chemical composition of argan oil. The authors selected argan tree samples from three locations, Ida Ou Semlal, Admin and Smimou, with distinct altitudinal levels, rainfall and temperatures. They found that the age of the trees and year of harvest did not significantly affect the fatty acid composition of the oils but confirmed the influence of geographical origin on fatty acid composition and that a specific correlation exists between fatty acids composition and individual tocopherols [17]. Unlike, Kharbach et al. [15], Aithammou et al. [17] did not attempt to show or apply multivariate statistics to prove that the fatty acid profile of argan could act as a determinant of provenance. 

Partially hydrogenated oils generally contain saturated and unsaturated fats, among them trans fats in variable proportions (with trans fats ranging from a few percent up to more than 50%), according to the production technology used. In cold-pressed vegetable oils, however, the amount of trans fatty acids is very low, generally between 0.1% and 0.3%. Olive oils are well characterised, and the range of fatty acids and the level of trans fatty acids is used to check the adulteration with refined oils [18]. Hilali et al. [19] found only traces of trans fatty acids (data not shown) in argan oil. This study deals with cosmetic argan oil, which, as for edible oil, the industry/consumer demands high quality, purity (unadulterated) and authentic argan products. Furthermore, unlike in previous studies, the full fatty acids profile, including, notably, the trans fatty acids, together with multivariate discriminant analysis to differentiate argan oil according to the origin, was used. In addition, this study used both linear discriminant analysis and latent structures discriminant analysis (OPLS-DA) since our previous studies found these methods to be much more powerful at classifying foodstuffs according to geographical origin [12,13]. 

Thus, the presented research aimed: (1) to characterise the fatty acid profile of argan oil including trans fatty acid isomers from five provinces in Morocco, namely Essaouira, Taroudant, Tiznit, Chtouka-Aït Baha and Sidi Ifni, and, thus, upgrade currently available data; and (2) to apply multivariate discriminant analyses to classify argan oil according to its geographical origin.

## 2. Results

Results on fatty acid composition together with the acidity and other relevant geoenvironmental parameters are collected in Appendix A. The acidity is one of the simplest but most important factors for determining the quality of oils. In this study, the acidity of the oil we tested ranged from 0.22% to 6.54%. Higher values indicate lower quality than food-grade oil but could still be used for cosmetics purposes. No statistically significant difference between acidity and the production region was observed. The literature data indicate that the acidity value of oil samples prepared from roasted nuts is consistently lower than those of oils prepared from non-roasted nuts [19]. All oil samples in our study were cold-pressed from unroasted kernels using mechanical extraction. 

Figure 1 shows the FA profile of the tested argan oils. The predominant fatty acid was oleic acid (C18:1) with the average content 46.6 ± 1.9%, followed by linoleic acid (C18:2; 32.6 ± 2.2%) and palmitic acid (C16:0; 13.1 ± 0.7%) (Figure 1). In argan oils, the prevailing fatty acids are monounsaturated (MUFA), followed by polyunsaturated (PUFA) and saturated fatty acids (SFA). 

Oleic acid content was in the range of 39.5–50.8% and linoleic acid in the range of 27.1–39.3%. The range of the sum of oleic and linoleic acid was narrower (76.4–81.3%) than other oils, and myristic acid (C14:0), palmitoleic acid (16:1), linolenic acid (C18:3), gadoleic acid (C20:1), behenic acid (C22:0), erucic acid (C22:1) and lignoceric acid (C24:0) were all <0.5%. Overall, the fatty acid profile and the data are comparable with other studies [15,20,21].

As a natural product, variability in the proportions of MUFA/PUFA/SFA is expected; this is especially true since our sample collection includes oils from multiple growing regions. Kouidri et al. [22,23] reported contents of MUFA (C16:1, C18:1 and C20:1) from 45.6% to 50.8% and PUFA (C18:2, C18:3) from 29.1% to 37.0%, in oils from two different regions. Beside geographical origin, the fatty acid distribution is also affected by the argan variety, year of harvest and harvest time [24]. For example, Aithammou et al. [17] reported a variation in UFA content between 78.3% and 81.8% (SFA between 18.2% and 21.7%). 

## 3. Discussion

Purity characteristics should also include the limits for total trans-oleic isomers (C18:1T) and total trans-linoleic + trans-linolenic isomers (C18:2 CT + C18:3 CTC). For example, the limit value for trans-oleic and trans-linoleic isomers in extra virgin and virgin olive oil is ≤0.05%, and, for total trans-linoleic + trans-linolenic isomers, the limit is also ≤0.05%. For refined olive oil and olive pomace oil, the limits are higher, ≤0.2% and 0.4%, respectively. The trans-fatty acid content in argan oils from different geographic areas is presented in Figure 2. 

The content of trans-oleic and trans-linoleic isomers are ≤0.05%, while trans-linolenic isomers are between 0.06% and 0.09%. Conversely, the sum of total trans-linoleic and trans-linolenic isomers exceeds the value of 0.05%. However, if we take into account uncertainty (expressed as an absolute value, k = 2) for trans C18:2 and trans C18:3 of 0.03, the results most likely comply with the limit values for virgin olive oils as laid out in Commission Regulation (EEC) No 2568/91 [18]. 

Statistical evaluation of the data indicates that the variance in the studied fatty acids was not significantly different (*p* > 0.05), and therefore an ANOVA was performed. First, we checked to see if the fatty acids contents were in agreement with the limit values for extra virgin argan oil. The most significant differences between the samples were in the amounts of palmitic, oleic and linoleic acid. Hence, we focused on determining the influence that the geographical origin has on the amounts of these three fatty acids—these also happen to be the most abundant fatty acids in argan oil. The 66 studied samples were divided into five groups, according to the region (province) of production (Table 1). Because the Lavene’s test showed no statistical difference between the variances of the studied samples, an ANOVA analysis was performed where Tukey’s HSD post-hoc and Duncan’s post-hoc tests were used to evaluate the ANOVA results. The Tukey test revealed a significant difference (*p* = 0.005) in the palmitic acid content between samples from Essaouira (12.7 ± 0.5%) and Sidi Ifni (13.8 ± 0.6%). A significant difference (*p* = 0.04) in the oleic acid content was also found between samples from Sidi Ifni (47.7 ± 2.1%) and Chtouka-Aït Baha (44.6 ± 1.2%), and the amount of linoleic acid in samples from Sidi Ifni (30.2 ± 2.6%) differed from that in samples from Essaouira (33.6 ± 2.4; *p* = 0.08) and Chtouka-Aït Baha (34.3 ± 1.1%; *p* = 0.01).

Discriminant analysis (DA) was further used to check if the fatty acid composition can discriminate between different regions of production. This calculation was performed on all samples from 2017 except for Agadir-Ida-Ou Tanane province (only one sample) and the unknown samples. Figure 3a shows a DA plot according to five production locations, while a discriminant loading plot is presented in Figure 3b. In a functional score plot, each group is represented by a scatter plot, while, in the loadings plot, a set of vectors indicating the degree of association of the corresponding initial variables with the first two discriminant functions. In the latter, the degree of distribution of each parameter in the classes is revealed. 

The groups Chtouka-Aït Baha, Essaouira and Tiznit provinces are well separated, while the other two groups overlap with these three groups. Function 1 explained 46.4% of the total variance with C24:0, C20:0 and C18:0 having the highest discriminating power. The critical parameters for Function 2, which explains 32.7% of the variance, were C16:1, C17:1 and C18:0. The group from Chtouka-Aït Baha province is well separated from the others due to the higher C18:0, C18:2, C18:3 and trans-fatty acids (Figure 2). For the Essaouira group (upper right part in the graph), C20:1, C17:1 and C22:1 content had the highest discriminating power. The prediction ability was the highest for Chtouka-Aït Baha province (90.0%) followed by the Essaouira (61.9%), while the lowest prediction ability was determined for the Sidi Ifni province (14.3%). The overall correct classification rate was 51.6%.

Furthermore, OPLS-DA tests for pairwise comparisons among all five regions (Figure 4 and Figure 5) was performed based on 13 significantly different fatty acids (*p* < 0.05), similar to the study performed by Chung et al. [25]. In the OPLS-DA models, leave-one-out was automatically used as cross-validation to obtain the misclassification result. The accuracy (CA), precision, recall and F1-score were calculated and are presented for each model in Figure 4 and Figure 5. The classification rate of these OPLS-DA models was between 82% and 100%, denoting the high quality and goodness of fit and predictability (≥0.92) to differentiate among the geographical regions in Morocco. The lowest prediction was obtained between the Sidi Ifni and Tiznit province and Essaouira and Taroudant provinces. Chtouka-Aït Baha province displayed the best model predictability, which supports our DA model. This province is the only one located in Sub-Sahara and is influenced by the arid Mediterranean climate conditions. The variable importance in the projection (VIP) values of OPLS-DA models are presented in Figure 4 and Figure 5. 

A VIP value of more than 1.0 revealed that the corresponding variable was important in discriminating geographical origin [25]. The C18:0 value is the main parameter for discriminating oil from the Chtouka-Aït Baha and Essaouira provinces (VIP value: 1.248–1.924) and oil from the Chtouka-Aït Baha and Taroudant provinces (VIP value: 0.603–2.144), while the levels of C16:1 can discriminate between oils from Chtouka-Aït Baha and Sidi Ifni and Chtouka-Aït Baha and Tiznit provinces (VIP values: 0.841–1.998 and 0.944–1.923, respectively). The province of Essaouira lies in an oceanic coastal region with hot humid and arid bio-climate, and the oil is distinguishable from that from Sidi Ifni and Tiznit provinces because of the discriminating power of C20:0. 

The Taroudant province is the furthest location from the coast and is surrounded by the High Atlas and Anti-Atlas mountain chains. It has a hot semi-arid bioclimate. This region’s oil can be separated from provinces of Tiznit and Essaouira based on its C24:0 content, while C20:0 has the most discriminant power for separating oils from this region from those of Sidi Ifni. Although no correlation between the content of fatty acid and temperature and amount of precipitation was observed, the variability of the annual precipitation and the bio-climatic conditions in the five regions can influence the fatty acid composition of argan oil [15]. If confirmed over the years, the variation in C18:0, C16:1, C20:0 and C24:0 could become useful markers to ascertain the geographical origin of argan oils.

## 4. Materials and Methods 

### 4.1. Sample Collection 

Seventy-three authentic samples of cosmetic argan oil were provided by CNESTEN (Centre National de l’Energie des Sciences et des Techniques Nucléaires (CNESTEN), Morocco). Sixty-six samples were collected from five separate provinces samples in the southwest of Morocco: Essaouira, Taroudant, Tiznit, Chtouka-Aït Baha and Sidi Ifni (Figure 6). Seven samples were of unknown origin (Table 1). Samples, harvested between July and August 2017, were collected from known co-operatives and with PGI certification in the field of oil extraction by NORMACERT (control and certification organism approved by the Ministry of Agriculture) [8]. 

All samples were for cosmetic purposes produced from unroasted kernels using mechanical extraction. The extraction process includes mechanical cleaning the nut from the pulp, manual cracking and seed separation from the stone, mechanical pressing and filtering the oil through a cellulose-plate filter [8]. Samples were placed in dark amber glass bottles, wrapped with aluminium foil and stored in the dark at room temperature until analysis. The exact time of harvesting is not known, except that they were collected during July–August 2017 when the argan fruits were fully ripe. However, the time of pressing is known, and these data are included in Appendix A.

### 4.2. Determination of Acidity 

Determination of free fatty acid content was carried out according to the Commission Regulation (EEC) No. 2568/91 based on the characteristics of olive oil and olive-residue oil and the relevant methods of analysis, last amended with Commission Implementing Regulation (EU) 2019/1604 [18]. Briefly, each sample was dissolved in a neutralised mixture diethyl ether:ethanol (1:1 *v/v*) and titrated with a 0.1 mol L*^−^*^1^ solution of potassium hydroxide. The indicator was phenolphthalein (10 g L*^−^*^1^, ethanolic solution). The content of free fatty acids is expressed as acidity calculated as the percentage of oleic acid.

### 4.3. Determination of Fatty acid Composition 

The fatty acid composition was determined out according to the EU method (Commission Regulation (EEC) No. 2568/91) [25]. Fatty acid methyl esters (FAME) were prepared with 2-M methanolic potassium hydroxide solution and the resulting FAME determined by gas chromatography with flame ionisation detection (Agilent Technologies, Santa Clara, CA, USA). The separation was achieved using a fused silica capillary column SP-2560 (100 m × 0.25 mm × 0.20 μm film thickness; Supelco Inc., Bellefonte, PA, USA), with hydrogen as the carrier gas. Samples were injected in the split mode (100:1) at an inlet pressure of 133 kPa. The temperature programme was as follows: 21 min at 180 °C; from 180 to 190 °C at 4 °C min^−1^; 11 min at 190 °C; from 190 to 220 °C at 10 °C min^−1^; 10 min at 220 °C; from 220 to 240 °C at 30 °C min^−1^; and 5 min at 240 °C. The FID temperature was 300 °C. Compound identification was based on matching performed peak retention times with those of standards FAME. The amount of an individual fatty acid is expressed as the percentage of FAME of the total fatty acids’ methyl esters.

### 4.4. Statistical Analysis 

Both Excel and SPSS programs (version 24) were used for data processing. A one-way ANOVA analysis of data was performed to study the influence of geographic origin on fatty acid composition, followed by post-hoc tests (Tukey’s and Duncan’s tests). Linear discriminant analysis (DA) using the XL-STAT software package (Addinsoft, Long Island, NY, USA, 2019) was used to determine the key factors responsible for determining the geographical origin. Moreover, orthogonal projections to latent structures discriminant analysis (OPLS-DA) was performed on all data to verify the geographical indication of argan oil samples.

## 5. Conclusions

Argan oil production economically supports millions of people in Morocco. To protect both producers of this valuable product and consumers from fraudulent activity, it is essential to have the necessary analytical tools for controlling the origin and the authenticity of argan oil. The fatty acid profile, combined with multivariate analysis, such as DA and OPLS-DA, was evaluated and used to classify argan oil samples according to its geographical origin. The overall correct classification rate was 51.6%. The separation was further improved using pairwise comparison by OPLS-DA to give an overall correct classification rate of 92%. Although the results obtained are promising, the robustness of the classification methodology requires further improvement by including many samples from different years of production. Despite this, we show that fatty acid composition could be used as a tool for verifying the provenance of argan oil. Furthermore, upgrading the database on the content of fatty acid trans-isomers can be used as a tool for detecting refined oils as well as a tool for good production practice. The method is easy and rapid and thus can be recommended for laboratories to control the authenticity of cosmetic argan products.

## Figures and Tables

**Figure 1 molecules-25-04080-f001:**
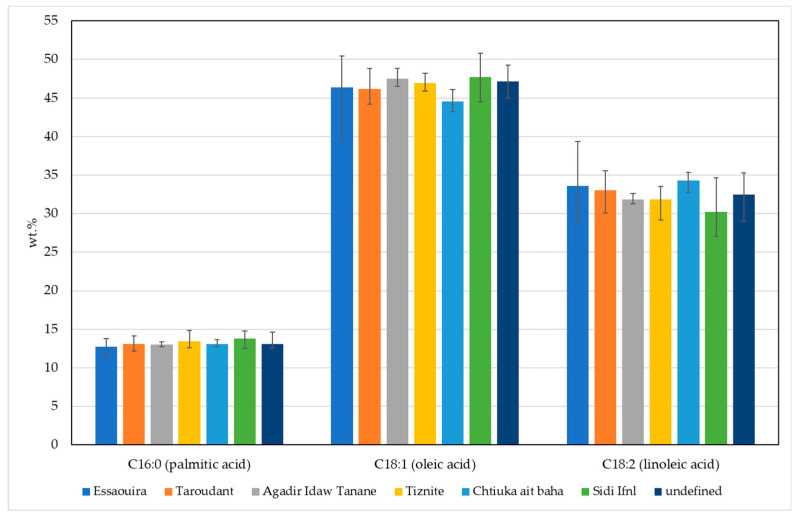
FA composition in argan oil: average palmitic (C16:0), oleic (C18:1) and linoleic (C18:2) acid content in argan oils from different geographic areas in Morocco. Minimum and maximum determined values in of each fatty acid in each area are presented with black lines.

**Figure 2 molecules-25-04080-f002:**
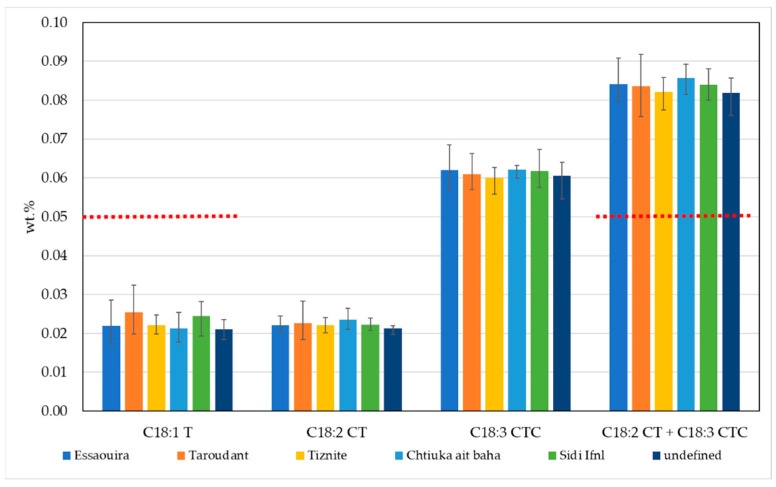
The trans-fatty acid content in argan oils from different geographic areas in Morocco. Red lines represent limit values for total trans-oleic isomers and total trans-linoleic + trans-linolenic isomers in extra virgin and virgin olive oil, according to the Commission Regulation (EEC) No 2568/91 [18].

**Figure 3 molecules-25-04080-f003:**
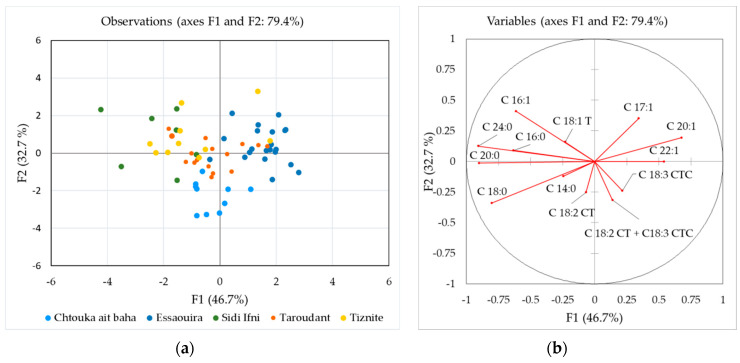
Discriminant analysis (DA) of the fatty acid composition according to the production region: (**a**) discriminant function score plot of authentic argan oil samples from Morocco 2017; and (**b**) discriminant loading plot showing correlations between initial variables and discriminant functions.

**Figure 4 molecules-25-04080-f004:**
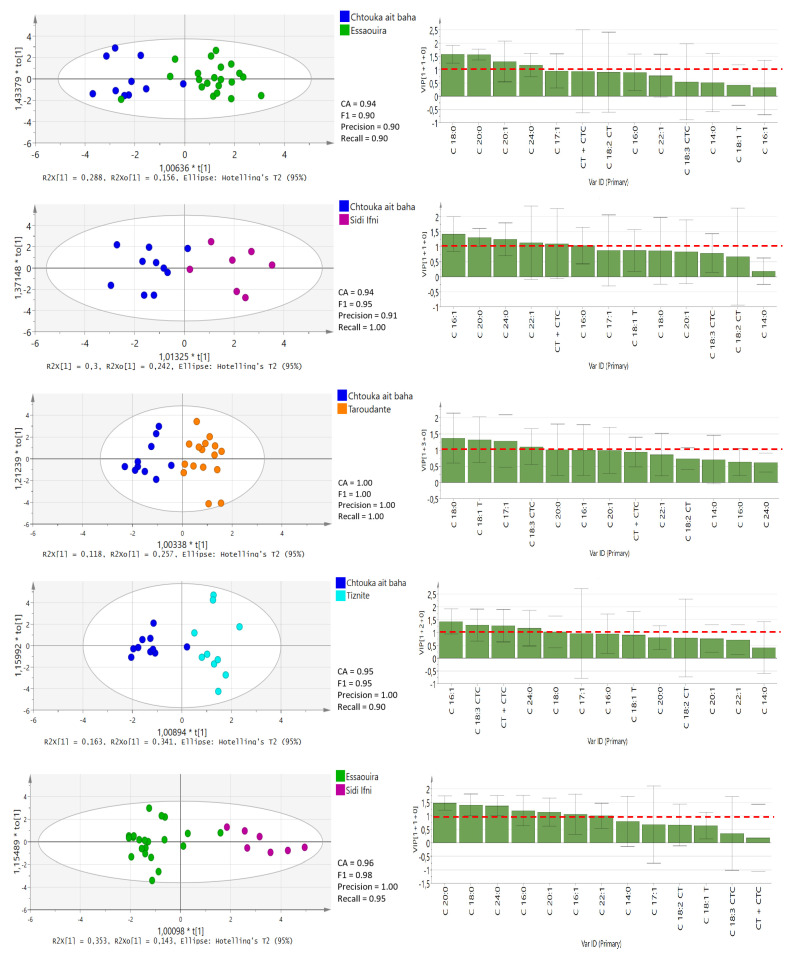
OPLS-DA score plots with VIP values in the pairwise comparison between different provinces derived from all fatty acid compositional data. The red-dotted line at VIP = 1.0 indicates criteria for the identification of the most important model variable. CT + CTC, total trans-linoleic + trans-linolenic isomers (C18:2 CT + C18:3 CTC).

**Figure 5 molecules-25-04080-f005:**
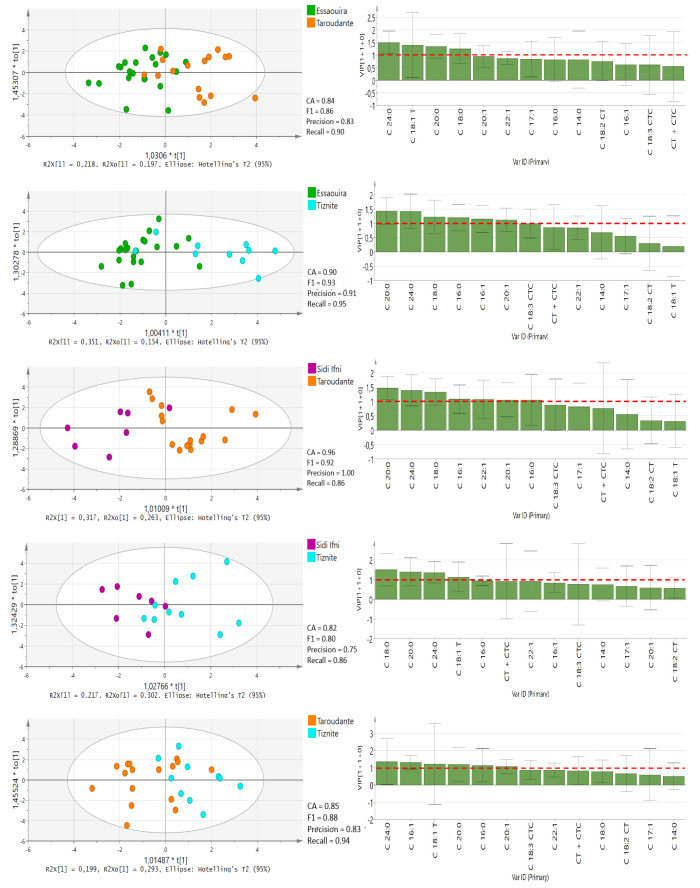
OPLS-DA score plots with VIP values in the pairwise comparison between different provinces derived from all fatty acid compositional data. The red-dotted line at VIP = 1.0 indicates criteria for the identification of the most important model variable. CT + CTC, total trans-linoleic + trans-linolenic isomers (C18:2 CT + C18:3 CTC).

**Figure 6 molecules-25-04080-f006:**
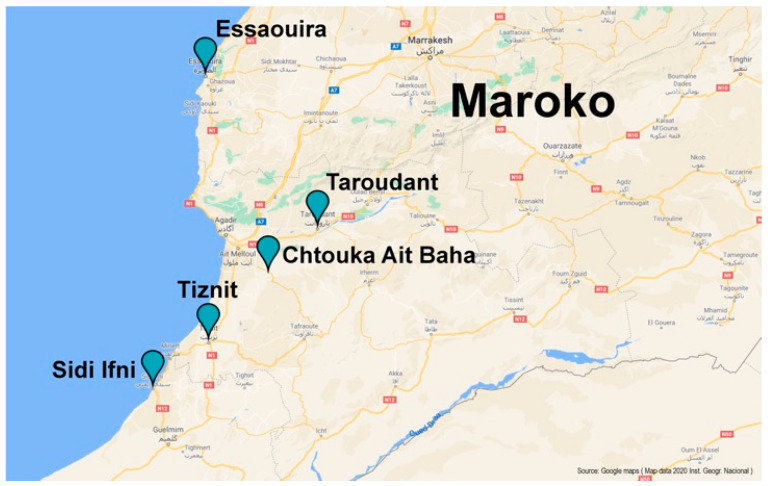
Different Moroccan provinces and corresponding districts (approximate location) from where argan seed samples were obtained.

**Table 1 molecules-25-04080-t001:** The origin of samples collected in 2017.

Region	N (Number of Samples)
Essaouira	23
Taroudant	16
Tiznit	10
Chtouka-Aït Baha	10
Sidi Ifni	7
Unknown origin	7
**Total**	**73**

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
