# Peer review of "Fatty Acid Composition of Cosmetic Argan Oil: Provenience and Authenticity Criteria"

_molecules, 2020, doi:10.3390/molecules25184080_

Round 1
Reviewer 1 Report
The paper entitled “Geographical discrimination of argan oil based on fatty acid composition” has been studied and evaluated by this reviewer.
The implementation of geographical indications tries to benefit rural economies by encouraging the traditional agricultural production linked to a specific geographical area. Thus, the development of analytical methods to check traceability and verify the geographical origin of the product holding this kind of recognition is of great interest to both scientific community and final consumers.
Overall, the manuscript is well organized. The Introduction is easy to read and gives a clear overview of what is argan oil and how it is produced; it also describes previous works in the field, citing the most important references. Moreover, the Results and Discussion sections present the outcomes of the study in a clear way. Nevertheless, there are some pending aspect that authors should clarify before I can recommend this contribution for publication in Molecules.
In the introduction, the novelty of the study should be better justified. Authors must point out what makes their contribution distinctive, specifically compared to some of the papers they previously cite (refs 14, 15, 16).
In my opinion sample selection should be better described and justified. Why did you include just one sample from Agadir-Ida ou Tanane? In the end, this sample had to be excluded from statistical analysis reducing the number of provinces within the models to 5. Moreover, authors state in line 130 “our sample collection includes oils harvested at different times”. Were all the samples coming from the same region harvested at the same time or did the authors include different time-points from each province? It would be interesting to include date of collection or any parameter related to fruit maturity in table S1 and confirm whether this parameter is responsible for sample grouping/separation in any way.
Authors are studying “cosmetic” samples (line 226) that are supposed to be produced from raw argan kernels. However, they check whether their acidity match Moroccan Normalisation guidelines for edible oils, and state that samples with higher acidity value “are of lower quality, but could still be used for cosmetics purposes”. Does this statement implies that the rest of the samples could be consumed as food? Otherwise, it would not make sense to check if a non edible oil meets the quality criteria included in guidelines for edible oils. However, to my understanding, the production of edible argan oil includes a roasting step that trigger several reactions that modify oil flavour (Maillard reaction, Strecker degradation and oxidation processes) and reduces the content of saponins, which are sometimes considered as toxic phytochemicals. In the same way, authors compare trans-fatty acids content to the limits given in EEC No 2568/91 for olive oil. Again, if the object of study in this work is cosmetic argan oil, why do authors evaluate a parameter established for an edible oil. As a potential reader, I would appreciate that authors clarify these aspects in the introductory section when describing the aim of the study.
Figures:
Figure 1 cannot be read. I would recommend to increase the size of the graphs to full page width. In addition, in my opinion, figure 1. a) is dispensable.
Figure 3 is named “Figure 1”; please, correct. I would also recommend to improve figure quality. (Size could be adjusted by moving part a legend to the bottom of the figure and expanding the graph; font size should also be enlarged)
Figures 4 and 5 are not legible. They are extremely blurred. In this case, apart from improving their sharpness, I would recommend to increase font size while keeping graphs proportions in the figure
Other comments:
- Lines 37-40: Long sentence, please rewrite it. Se second part of the sentence is not well connected.
- Argan is sometimes capitalized; please, unify criteria.
- Line 48: Please, correct “Ref [1,1]”
- Line 65: Pease, include a reference to the applicable legislation.
- Line 71: Sentence structure in “a method based on triacylglycerol profiles high-performance liquid chromatography with evaporative light scattering detection (HPLC-ELSD)” is not correct. Please, rewrite.
- Line 77: Please, modify word order: “which could be used” must be together.
- Line 83: Please, add a comma after “provenance”
- Lines 108, 109. Fatty acid composition and acidity are presented in Table S2; please correct the reference to Supplemetary Materials.
- Line 253: please, remove the word “performed”
- Line 278: Shouldn´t it be “provenance OF argan oil”?
Author Response
See attachement.

Reviewer 2 Report
The article is interesting as they carry out a methodology to determine the quality of argan oil.Only a few small errors were found: 1) page 3, line 109, says Table S1 but the correct is Table S2
2) page 5, line 177, says Figure 1 but the correct is Figure 3
3) in lines 250 and 251 put -1 as superscrip
4) It was very difficult to see the letters in Figures 1,4 and 5, they are not sharp at all. They must be improved
5) There is no indication in the document regarding figure 3b
Author Response
See attachement.

Reviewer 3 Report
GENERAL COMMENTS
The manuscript “molecules-887000” entitled “Geographical discrimination of argan oil based on fatty acid composition” is a very good work. Only few corrections are recommended to improve the quality from the point of view processing and presentation of data.
I recommend to publish the paper after the following minor corrections.
CONCEPTUAL ERRORS
Figure 1a: in every scores plot, it is recommended to use the same scale from abscissa and ordinate; in this case, for instance, they could be both from -7 to 5. The reason is that zooming in or zooming out one of the two axes produces a distortion of the real distance between clusters.
Figures 4 and 5: the scores plots on the left should be adjusted with respect to the scales (see comments about Fig. 1a). Moreover, all the scores plot in these figures should be represented without Hotelling ellipses. In fact, Hotelling ellipses make sense only when a homogeneous class is projected (for instance: argan oils from the same region; even all the samples of tables S1 and S2, but only if the category is “argan oil” to be compared with “other oils”). Indeed, the Hotelling ellipses reported in figures 4 and 5 are not relevant for the discussion referred to this figure. As a consequence of this correction, the following correction is needed in captions to figures 4 and 5:
Lines 210-211 and 216-217. ERRATA “The ellipse on the score plot represents the 95% confidence interval, and the red-dotted line at VIP=1.0 indicates criteria for the identification of the most important model variable.” CORRIGE “The red-dotted line at VIP=1.0 indicates criteria for the identification of the most important model variable.”
EDITING ERRORS
Line 36: ERRATA: “more._It is” CORRIGE: “more. It is”
Line 39: ERRATA: “) represents” CORRIGE: “), representing”
Line 77: ERRATA: “which could, in addition to measuring the campesterol content, be used to detect adulteration.” CORRIGE: “which could be used to detect adulteration, in addition to measuring the campesterol content.”
Line 83: ERRATA: “provenance other” CORRIGE: “provenance, other”
Line 95: ERRATA: “and, in some cases preparation method” CORRIGE: “and, in some cases, according to the preparation method”
Line 108: ERRATA: “in Table S1” CORRIGE: “in Tables S1 and S2”
Lines 169 and 173: ERRATA: “most” CORRIGE: “highest”
Line 170: ERRATA: “The parameters important” CORRIGE: “The most important parameters”
Figure 4 and 5: the quality of plot is very poor. Please improve their graphic resolution.
Line 202: ERRATA: “discriminatory” CORRIGE: “dicriminant”
Lines 250-252 . ERRATA: “min−1” CORRIGE: put “-1” in superscript
Author Response
See attachement.

Round 2
Reviewer 1 Report
The revised version of the paper entitled “Geographical discrimination of argan oil based on fatty acid composition” together with the authors response to my previous comments have been evaluated by this reviewer.
I appreciate the improvement on figures’ quality and minor corrections that the authors have done. However, my main concerns remain unsolved. I may not made myself clear enough, but the aspects they didn´t address make me consider that the paper does not make a substantial contribution to the start of the art and thus, it does not deserve publication in Molecules.
I would be happy to reconsider my decision if authors made the necessary changes in the manuscript (not only in the response to the reviewer comments), to clarify the following aspects:
1) In the introduction, the novelty of the study should be better justified. Authors must point out what makes their contribution distinctive, specifically compared to some of the papers they previously cite (refs 14, 15, 16).
It has not been done. The info they added at the end of the introduction is not a justification of the novelty of their work. In fact, again, they reference a regulation for olive oil, which does not completely make sense to me (see comment number 3).
2) In my opinion sample selection should be better described and justified. Why did you include just one sample from Agadir-Ida ou Tanane? In the end, this sample had to be excluded from statistical analysis reducing the number of provinces within the models to 5. Moreover, authors state in line 130 “our sample collection includes oils harvested at different times”. Were all the samples coming from the same region harvested at the same time or did the authors include different time-points from each province? It would be interesting to include date of collection or any parameter related to fruit maturity in table S1 and confirm whether this parameter is responsible for sample grouping/separation in any way.
Authors decided to remove the sample from Agadir-Ida ou Tanane from Figures and statistics, but they keep talking about 6 regions in the abstract and when describing the aim of the study in the introduction. To my understanding, this is a bit misleading.
Saying that samples were collected between July and August does not give information regarding the possible effect of fruits maturity on samples classification. Were all the samples from the same region collected at the same time? Was the ripening degree the same one for samples coming from different regions? If they include the sentence “our sample collection includes oils harvested at different times”, as a reader, I expect to have the details of harvesting times and a discussion on how they can affect the composition (and groping) of the samples. The establishment of meaningful statistical models relies on the consideration of, at least, the variables that can have a greatest effect on samples differentiation. This discussion needs to be added to the manuscript.
3) Authors are studying “cosmetic” samples (line 226) that are supposed to be produced from raw argan kernels. However, they check whether their acidity match Moroccan Normalisation guidelines for edible oils, and state that samples with higher acidity value “are of lower quality, but could still be used for cosmetics purposes”. Does this statement implies that the rest of the samples could be consumed as food? Otherwise, it would not make sense to check if a non edible oil meets the quality criteria included in guidelines for edible oils […]. In the same way, authors compare trans-fatty acids content to the limits given in EEC No 2568/91 for olive oil. Again, if the object of study in this work is cosmetic argan oil, why do authors evaluate a parameter established for an edible oil. As a potential reader, I would appreciate that authors clarify these aspects in the introductory section when describing the aim of the study.
Authors state that they agree with my comments and that “Our intention was not to check if the samples correspond to quality parameters for edible oil”, but they keep this sentences in the manuscript “However, we should mention that > 87% of the samples have an acidity below 0.80% required by the Moroccan Normalisation guidelines for edible oils [17]. The remaining samples are of lower quality, but could still be used for cosmetics purposes.” They haven´t appropriate addressed my concern. They need to made several changes to clarify these aspects in the text of the manuscript.
Other comments:
Line 65 (line 72 in the revised version): Please, include a reference to the applicable legislation.
Author Response
See attachement.

Round 3
Reviewer 1 Report
The manuscript has been notably improved with the modifications made by the authors. Therefore, I think that the paper is now suitable for publication in its present form.